# Gamete Collection, Artificial Fertilization and Captive-Rearing of Eggs in a Terrestrial-Breeding Anuran with Parental Care: *Alytes obstetricans*

**DOI:** 10.3390/ani13172802

**Published:** 2023-09-04

**Authors:** Lucía Arregui, Jaime Bosch

**Affiliations:** 1Centro de Investigación, Seguimiento y Evaluación, Parque Nacional de la Sierra de Guadarrama, Ctra. M-604, Km 27.6, Rascafria, 28740 Madrid, Spain; jaime.bosch@csic.es; 2IMIB-Biodiversity Research Institute, University of Oviedo-CSIC-Principality of Asturias, C/Gonzalo Gutiérrez Quirón S/N, Mieres, 33600 Asturias, Spain

**Keywords:** amphibian, midwife toad, egg clutch, hatching, sperm, spermiation, ovulation, IVF, in vitro fertilization

## Abstract

**Simple Summary:**

Amphibians are the most threatened vertebrate class on Earth, and assisted reproductive technologies are essential tools to complement other conservation efforts. We established initial assisted reproductive technologies for *Alytes obstetricans*, a terrestrial breeding anuran whose males carry the egg strings twined around their hind limbs until hatching. Non-lethal gamete collection by injection of hormones was performed and allowed the collection of hormonally-induced sperm and eggs. However, poor response to hormonal stimulation of spermiation was obtained (range 17 to 50% of treated males). These gametes were combined and fertilization took place and completed embryo development in 10% of eggs. In addition, two methods for captive-rearing of embryos were tested: keeping the eggs together (mimicking natural arrangement) or splitting the eggs (to reduce fungal propagation) on moist paper. When eggs were separated, all clutches exhibited embryo survival (ranging from 27 to 61%) while only 8% of clutches presented embryo survival when eggs were kept together. This method for captive-rearing of eggs could be implemented in captive breeding programs for different species of *Alytes*. Although successful, sperm collection and artificial fertilization should be optimized to contribute to the management of wild and captive populations of this and related species.

**Abstract:**

The genus *Alytes* consists of six primitive terrestrial-breeding species (five of them endangered or present in restricted distributions) with unique male parental care. *Alytes obstetricans* was used as a model for the development of assisted reproductive technologies for gamete collection, artificial fertilization and captive-rearing of embryos. Sperm collection was achieved with human chorionic gonadotropin (hCG), but only 17 to 50% of males responded to the hormone. In addition, an effect of captivity on sperm quality was observed. A combination of gonadotropin-releasing hormone agonist (GnRHa) and hCG was tested for induction of ovulation, and 100% of females responded to the treatment. Recently collected sperm was used in artificial fertilization, and 10% of eggs showed complete embryo development. To design a method for captive-rearing of embryos, natural clutches (*n* = 13) were removed from males and divided into two sub-clutches. One was kept with all eggs together and in the other, the egg string was cut and eggs were split individually. All sub-clutches with eggs kept separated presented embryo survival (ranging from 27 to 61%) that hatched normally, while only one sub-clutch with embryos maintained together had some embryo survival (36%). These results may contribute to the management of this and related species.

## 1. Introduction

Assisted reproductive technologies are essential tools to complement in situ conservation efforts and to circumvent problems during captive breeding. Some progress in the development of these technologies for amphibians has been accomplished during the last decade [1], but considering the magnitude and global scope of the amphibian biodiversity crisis [2,3], the lack of knowledge about most amphibians’ reproductive biology compromises the success of these conservation efforts.

The genus *Alytes* consists of six primitive species endemic to southwestern Europe and northwestern Africa, commonly known as midwife toads. They are unique among amphibians in their parental care system; after terrestrial egg fertilization, males coil the strings of eggs around their hind limbs and carry them on land until embryonic development is completed. Then, males release the larvae into suitable water bodies. Three of these species, *Alytes dickhilleni, A. maurus* and *A. muletensis*, are endangered and present very restricted ranges in Spain or Morocco [4]. *Alytes cisternasii* is native to Spain and Portugal and *A. almogaravii* [5] to Spain and France, and their area of distribution is also restricted. *Alytes obstetricans* has the largest geographic range of all species, extending from the central Iberian Peninsula to the Alps, Germany and central Belgium, but some of its southern populations in Portugal are extinct [4]. The populations of these species are decreasing and severely threatened by the loss of suitable habitat and breeding sites, introduced predators [6] and chytridiomycosis and other diseases [7,8,9,10]. Ex situ populations kept in captive-breeding facilities could help to maintain the wild populations through the production of animals for reintroduction or reinforcement and increasing the knowledge of the biology of the species.

*Alytes* species have been successfully bred in captivity. *Alytes muletensis* was near extinction when in 1985 a captive-breeding program was established in the Channel Island Jersey Zoo (UK) by the Durrell Wildlife Conservation Trust. Other centers in the UK and Spain also bred this species, and releases of captive-bred specimens have been taking place since 1989 [11]. The population of this species is slowly increasing thanks to the investment in captive-breeding programs, reintroduction efforts and other conservation initiatives such as the control of invasive snakes [12] and *Chytridium* [13,14]. *Alytes dickhilleni* was described in 1995, and due to its very small distribution area, a captive breeding center was initiated in 2008 at Bioparc Fuengirola (Malaga, Spain), which received its first animals in 2010. In 1999, chytridiomycosis was found in a population of *A. obstetricans* in Guadarrana National Park (Madrid, Spain) [7] and a captive breeding center, The Captive Breeding Centre for Threatened Amphibians of Guadarrama Mountains, was founded for this species in 2008 to mitigate the negative effect of the fungus by releasing captive-reared larvae and wild larvae after treatment against the chytrid for reinforcing the natural population. A best practice guideline for captive maintenance of species belonging to this genus and a comprehensive book have been published, compiling the knowledge gathered from captive and wild populations [6,15]. To our knowledge, no captive populations of *A. maurus* or *A. cisternasii* have been established so far.

Despite the increase in knowledge and the captive breeding centers, the risk of losing any of these species is still very high due to their restricted distribution and decreasing population trends. In addition, climate change will affect the current distribution of species, and several models show a 40 to 88% potential distribution range constriction in Spain for some *Alytes* species for the next 40 years [16]. Assisted reproductive technologies have the potential to advance and complement current conservation efforts. Techniques for captive-rearing embryos, gamete collection and artificial fertilization (AF) allow rearing of clutches abandoned by males or that have to be removed from males’ limbs [15], interchange of genetic material between captive and wild populations without moving animals, and circumvent the lack of breeding between specific specimens. In addition, assisted reproductive technologies, together with germplasm banking through the cryopreservation of gametes, allow safeguarding of current genetic diversity.

*Alytes* egg clutches should be kept on wet paper instead of in water, a method that was developed at the Centro de Cría en Cautividad de Anfibios Amenazados de la Sierra de Guadarrama [6]. When abandoned egg clutches are maintained together on wet paper mimicking natural conditions, a low successful rate of development was found [6]. The effect of keeping the embryos together or separated to reduce fungal contamination during the developmental period has never been analyzed. In addition, gonadotropin hormones (mainly gonadotropin-releasing hormone (GnRH)) have been used to obtain ethograms for courtship and mating behavior in *Alytes* [17], and an unreliable result was obtained after an attempt for hormonal stimulation of mating [18]. Therefore, non-invasive gamete collection has never been tested in *Alytes*.

Thus, the goal of this study was to establish initial methodologies that facilitate future assisted reproductive technologies development for *Alytes* species. Specific objectives were to: (1) assess the effect of keeping the embryo together versus separated on embryo development during artificial rearing, (2) evaluate the feasibility of sperm collection by hormonal treatment, (3) analyze the effect of captivity in the response to hormonal stimulation of spermiation and (4) characterize the response of females to hormonal induction of ovulation and explore the feasibility of artificial fertilization in *Alytes obstetricans*. These results can be used as a model for species in this genus and for other terrestrial anurans.

## 2. Materials and Methods

### 2.1. Captive Animals

Captive animals (*Alytes obstetricans pertinax* [19]) were kept at The Captive Breeding Centre for Threatened Amphibians of Guadarrama Mountains located in Rascafria (Madrid, Spain). Animals were collected as tadpoles or were bred in captivity and reared until adulthood. Adult animals were kept in plastic containers with a gravel layer covered with pebbles and retreated sites provided. Animals were housed in bigger containers (42 cm L × 27 cm W × 16 cm D) in mixed-gender groups for the experiment involving natural mating (Experiment 1) and in single-sex female groups for hormonal induction of ovulation (Experiment 4). For hormonal stimulation of spermiation (Experiments 2 and 3) males were housed individually in smaller containers (24 cm L × 17 cm W × 16 cm D). Water for bathing was provided ad libitum, and an automatic irrigation system was established for two 7 min periods (at 09:00 and 21:00 h) daily. Animals were held at 19 °C and 12 h of light (09:00 to 21:00 h) and 12 h of darkness (21:00 to 09:00 h). Illumination was supplied by white fluorescent light and controlled by an automatic time switch. Animals were fed crickets dusted with calcium powder once a week.

### 2.2. Wild Animals

Wild males (*Alytes obstetricans obstetricans*) were caught around Ercina Lake (Asturias, Spain) in March 2021 with permits from Consejería de Medio Rural of the Principality of Asturias (permit ref. CO/09/033/2021). Animals were captured by hand during night and housed in Styrofoam containers with humid chicken-egg cartons that were also used as retreat sites. Animals were fed crickets dusted with calcium powder once a week. Toads were released into the wild in the same location two months after being captured after testing negative for *Batrachochytrium dendrobatidis* and *Ranavirus* by using standard qPCR analyses.

### 2.3. Experiment 1: Captive-Rearing of Embryos

To assess the effect of keeping the eggs together or split, egg clutches (*n* = 13) were removed by hand from the hind limbs of males when found. Each clutch was divided into two sub-clutches with scissors. One sub-clutch was kept with all eggs together while for the other sub-clutch, each egg was separated by cutting the string with scissors. Split eggs were kept 1–2 cm apart from each other (Figure 1a). All eggs were kept under indirect natural light, at room temperature (18 to 20 °C) on moist paper in a shallow container, and one side of the moist paper was submerged in dechlorinated tap water. Dead eggs were not removed. Eggs were kept until hatching, and hatched and non-hatched embryos were quantified.

### 2.4. Experiment 2: Hormonal Stimulation of Spermiation in Captive and Wild Males

To stimulate spermiation, two doses of human chorionic gonadotropin (hCG, Sigma, Madrid, Spain) were tested based on results from pilot trials performed with the same captive males during previous months (September to November 2020, Table 1). Urine was collected prior to hCG administration to verify the absence of spermatozoa. Males were weighed and were administered either an intraperitoneal injection of 15 IU hCG per gram of body weight (IU hCG/g) or an intranasal dose of 0.04 IU hCG/g prepared in phosphate-buffered saline (PBS, Sigma). For intranasal application, a drop of maximum volume of 2 µL was released in the opening of each nostril and was taken up during inspiration. After hormonal treatment, each male was placed in an individual plastic container on a moist paper at room temperature (16–18 °C for captive animals and 20–21 °C for wild animals). Urine was collected every 2 h post-administration at 2, 4, 6, 8, 10 and 12 h with a follow-up collection at 24 h post-hormone administration. Samples were obtained using a flexible vinyl catheter (outside diameter 1.32 mm; Scientific Commodities Inc., Lake Havasu City, AZ, USA) that was gently introduced in the cloaca to drain the urine. Males kept at the captive breeding center (*n* = 12) were randomly treated with both protocols separated for 7 weeks during February and April 2021. Males collected in the wild (*n* = 8) were split between both treatments; half of them (*n* = 4) received the intraperitoneal injection and the other half (*n* = 4) the intranasal treatment in April 2021 (16 days after being captured).

The volume of urine collected was measured using a pipette, and all samples were assessed for the presence of spermatozoa at a 400× magnification using a Nikon E600 microscope (Minato City, Tokyo, Japan). The sperm concentration was estimated using a hemocytometer. For evaluation of motility, samples with at least 10 sperm cells found were classified as displaying forward movement, movement of the flagella but stationary, or non-motile. The total motile sperm was calculated as the addition of sperm moving forward and showing movement of flagella only. These sperm cells were also assessed for morphology as presenting normal or abnormal head and normal flagella, round-ended flagella or other abnormal flagella. Osmolality of some urine samples (with and without sperm) was measured using a single-sample osmometer (Osmotech, Germany).

### 2.5. Experiment 3: Hormonal Stimulation of Spermiation in Captive Males after Semi-Free-Ranging Period

Captive males were released into a semi-free outdoor area at the captive breeding center on May 2021. This area measures 5 m L × 2.5 m W and it is surrounded by a solid metal fence that measures 150 cm height above- and belowground to prevent animals from escaping. Bricks and ceramic roof tiles were available as retreat sites, and water was available ad libitum in a 70 cm L × 60 cm W artificial pond (Appendix A). One month later, males (*n* = 14) were captured and housed individually inside the captive center. One day after capturing, a urine sample was collected before the hormonal treatment, and males were administered either 15 IU hCG/g or 20 IU hCG/g in PBS intraperitoneally. Urine samples were obtained every 3 h post-administration at 3, 6, 9 and 12 h with a follow-up collection at 24 h post-hormone treatment following the method described in the previous experiment. Evaluation of sperm motility was performed as described above.

### 2.6. Experiment 4: Hormonal Stimulation of Ovulation and Artificial Fertilization

For hormonal stimulation of ovulation, gravid females (with eggs visible through the abdomen, *n* = 5, Figure 2) were treated with an intraperitoneal ovulatory dose of 15 IU hCG + 0.8 µg [des-Gly10, D-Ala6]-LH-RH ethylamine acetate salt hydrate (GnRHa, Glass Chemicals, Leganes, Madrid, Spain) per gram of body weight and kept overnight in individual plastic containers on a moist paper at room temperature. After 11 to 13 h, eggs were collected by opening the cloacal sphincter with a standard microbiology inoculation needle (diameter 2 mm) and applying gentle pressure on the abdomen. Once some eggs were expressed, the egg string was cut with scissors to allow performing two or three artificial fertilization (AF; combination of sperm and eggs in a petri dish for species presenting external fertilization [20]) trials with eggs from three of the females. All eggs were collected together and fertilized with one sperm sample for the other two females. Sperm samples from wild males obtained during Experiment 2 were used to fertilize these eggs. An aliquot of sperm samples collected from the same male and obtained 4, 6 and 10 h post-hormonal treatment was kept in a refrigerator (they were stored for 22, 17 and 14.5 h) and used to fertilize 28, 29 and 7 eggs from three different females using 55, 80 and 40 µL of spermic urine. Samples from the male releasing sperm 24 h after hormonal treatment were used for another AF. Eggs from three females (6, 6 and 7 eggs) were fertilized with 50 µL of spermic urine obtained 5 to 10 min before and kept at room temperature (21 °C). Finally, eggs from these same three females (15, 15 and 14 eggs) were fertilized with 15, 20 and 30 µL of spermic urine less than 2 min after collection. To perform AF, eggs were expressed on the center of petri dishes, sperm was pipetted on top of them and after 3 to 12 min, eggs were transferred to humid paper. Alternatively, eggs were directly expressed on humid paper and sperm was added. After 3 to 4 h, the string between the eggs was cut and eggs were separated to be kept as described in Experiment 1. Humid paper and water were changed every three days after AF. Broken eggs were removed daily. Seven days after AF, the number of live embryos was counted as those presenting pigmented optic discs (Figure 1b). After AF, an aliquot of spermic urine was obtained from the male to assess sperm concentration and motility.

### 2.7. Statistical Analysis

To analyze the differences in the percentage of hatched embryos between eggs kept together or split (Experiment 1), a Wilcoxon test was used and performed with JMP Pro 17 (SAS Institute Inc., Cary, NC, USA). To examine the proportion of males responding to the hormone treatments, two analyses were performed using a binary model with IBM SPSS 15 for Windows (SPSS Inc., Chicago, IL, USA). A generalized estimating equation was used to analyze the effect of the route of hormone administration on captive males (Experiment 2). A generalized linear model was used to compare between the two doses of hCG for males after a semi-free-ranging period (Experiment 3). Animal weights and volume of urine were compared by *t*-test and Mann–Whitney *U*-test, and normality was tested with a Shapiro–Wilk test. Because there were low responses during spermiation trials, only descriptive statistics are reported for sperm quality variables. Data are expressed as mean ± SD, and *p* < 0.05 was considered significant.

## 3. Results

### 3.1. Experiment 1: Captive-Rearing of Embryos

The total number of carried eggs was 58.0 ± 14.5 per male. The mean number of eggs kept in the sub-clutches with eggs together was 23.0 ± 6.0 while 35.1 ± 10.1 were maintained in split sub-clutches. Keeping eggs together had a significant and negative effect on embryo survival (S = −45.5, *t*-ratio = −10.3939, *p* = 0.001). Only one out of the 13 sub-clutches with all eggs together had some embryo survival (36%), while all sub-clutches whose eggs were kept separated presented some survival (44.1 ± 10.1%, range 27–61%).

### 3.2. Experiment 2: Hormonal Stimulation of Spermiation in Captive and Wild Males

None of the males had sperm before the hormonal treatment. The volume of urine obtained was 43 ± 41 µL (ranging from 2 to 250 µL, *n* = 241). Osmolality of urine without sperm was 29.8 ± 11.3 mOsm/kg (*n* = 8). The model showed no effect of the route of hormone administration (intraperitoneal vs. nasal) on the proportion of responding males in captivity (*p* > 0.05). In captive males (*n* = 12), sperm was observed in six males treated intraperitoneally with hCG, but in two of them, only few sperm (2–7 spermatozoa out of 20 fields) were found and in a single time point. Therefore, only four individuals presented sperm at several and successive time points and were considered as responding males (Table 2). Sperm concentration from these samples ranged from 1.8 × 10^6^ to 5000 sperm/mL. Intranasal treatment stimulated spermiation in two males that also responded to intraperitoneal treatment. Sperm concentration ranged from 0.2 × 10^6^ to 12,500 sperm/mL. Slow forward motility was found in <7% of the sperm cells analyzed (the number of evaluated cells ranged from 10 to 51 sperm cells per sample), and total motility was 43 ± 26%. Many abnormal sperm were found, with 49% of sperm presenting normal head and 24% with normal flagella. Flagella showing the last part coiled was the most common abnormality, with close to 65% of sperm presenting this feature. None of the wild males (*n* = 4) responded to intranasal treatment, and two of them released sperm after intraperitoneal injection. Sperm concentration from these samples ranged from 0.2 × 10^6^ to 5000 sperm/mL. Similarly, 8% of sperm presented forward progression (the number of evaluated cells ranged from 16 to 73 sperm cells/sample), but faster and straight movement was observed in samples from one male. Percentage of total motility was 68 ± 21%. Sperm presenting normal head was 71% and normal flagella 28%. Sperm with coiled flagella was 64%. Urine volume of samples presenting sperm was lower in captive males (28.1 ± 27.7 µL, range 5 to 80) than in wild males (59.3 ± 25.9, range 15 to 90, Z = −2.198, *p* = 0.03), although captive males were heavier (6.30 ± 1.11 g) than wild males (3.60 ± 0.67 g, *p* < 0.001). Morphological differences among subspecies of *Alytes obstetricans* have been previously described [19].

### 3.3. Experiment 3: Hormonal Stimulation of Spermiation in Captive Males after Semi-Free-Ranging Period

Sperm were not found in the urine of any male before the hormonal treatment. The volume of urine sample obtained was 80 ± 73 µL (ranging from 2 to 350 µL, *n* = 85). There were no differences in the male weight between both groups (7.3 ± 1.4 vs. 7.1 ± 1.2 g, *p* > 0.05). Treatment with 15 IU hCG stimulated spermiation in three males, and two males responded to 20 IU hCG out of seven in each group (Table 3). The model showed no difference in the percentage of responding males between hormone doses (*p* > 0.05). Sperm concentration over time was highly variable among males in both treatments (Table 3). The percentage of sperm showing forward progression was low (4%, evaluated cells ranged from 61 to 100 sperm/sample) and total motility was 46 ± 19%. Osmolality of urine presenting sperm was 73.5 ± 10.7 mOsm/kg (*n* = 4).

### 3.4. Experiment 4: Hormonal Stimulation of Ovulation and Artificial Fertilization

After hormonal stimulation of ovulation, eggs were collected from all females (26 ± 3 eggs/female, range 22 to 29, *n* = 5). Embryo development was not observed in eggs fertilized with refrigerated or room-temperature stored sperm. Some eggs (2 out of 15 and 1 out of 14) from two of three females fertilized with sperm immediately after collection were alive seven days after AF (Figure 1b). These embryos were moved to clean humid paper and kept under the same conditions. Three normal tadpoles hatched between 16 and 17 days after AF (Figure 1c). Sperm concentration measured after this AF was 0.7 × 10^5^ sperm/mL, forward motility was 6%, and total motility 13%. After collection, eggs were still visible through the skin in at least one of the females. Nine days after AF, all females were checked for egg presence, and eggs were observed in four of the females on at least one side of the body.

## 4. Discussion

Amphibians present greater reproductive modes than any other vertebrate [21,22], so developing generalized methods for assisted reproductive technologies is challenging. This study provides the first investigation on these technologies in a species with terrestrial fertilization and parental care as a model for other species.

*Alytes* has been successfully bred in captivity since 1988 (*A. muletensis*) [18], but terrestrial parental care complicates captive-rearing of embryos compared with most anurans. Occasionally, wild or captive males carrying eggs have been found to release and abandon the clutch if handled, as a result of inexperience of the male, interference from other individuals or for unknown reasons [6]. In addition, sometimes eggs have to be removed from the male’s limbs when constriction of the limb takes place [15] or for other reasons (e.g., egg strand improperly attached to the male’s legs or to force a new amplexus [6]). A reliable method for captive-rearing of clutches was still needed because these embryos died when kept in water [23]. Our results show that cutting the string, separating the embryos and keeping them on a permanently humid paper allows complete embryo development. The low success of embryo development when embryos were kept together (>90% of clutches lost all embryos) contrasted with the 100% of clutches presenting some embryo survival when eggs were separated. Mean hatched eggs for this species in a wild population was 74% [24], so this method of artificial rearing can lead to between 36 and 82% of the natural hatching rate. Probably, separation avoids fungus transmission among embryos that, in nature, could be controlled by skin secretion from the carrying male. This method could be incorporated as the best practice for rearing embryos in captivity when necessary.

The first assisted reproductive technique to be developed for contributing to the conservation of a species should be the non-lethal collection of sperm. Hormonal stimulation of spermiation was feasible in *Alytes obstetricans*, but few males responded to the treatment. The current protocol needs to be adjusted for stimulation of spermiation testing a broader range of concentrations, other hormones such as follicle-stimulating hormone [25] or a combination with a dopamine antagonist [26]. Although not many males released sperm after the hormonal treatments, several patterns were observed:(1)The spermiation response using hCG was more effective than using GnRHa. GnRHa was not able to stimulate spermiation at any of the concentrations and routes tested. However, we cannot dismiss the possibility that the concentration of GnRHa administrated exceeds the optimum for this species. hCG and GnRHa have different routes of action that are highly conserved among vertebrates. GnRHa is a synthetic analog of GnRH that is released by the hypothalamus and stimulates the production of luteinizing hormone (LH) and follicle-stimulating hormone (FSH) by the pituitary gland. LH and FSH act in the testes to stimulate testosterone release by Leydig cells and control spermatogenesis through the Sertoli cells, respectively. hCG binds to the same receptors as LH on the Leydig cells [27], and these hormones are involved in spermiation (release of sperm attached to the Sertoli cells) through their action on the Sertoli cells in anurans [28,29]. Species belonging to different families in anurans have shown differences in the sperm-release response triggered by these exogenous hormones [30,31,32,33], but further studies are required to clarify the mechanisms.(2)No differences in the proportion of responding males were found between intraperitoneal injection and intranasal application of hCG in captive males. Hormone administration in amphibians is commonly performed intraperitoneally or subcutaneously in the dorsal lymph sacs, but alternative non-invasive routes have been tested. Hormones have been administrated topically [34,35,36] and through intranasal application [37]. Although intraperitoneal treatment and intranasal application induced spermiation in 33% and 17% of males, respectively, it would be desirable to develop a reliable non-invasive method for hormonal treatment because it is less invasive for animals and it is easier to perform requiring less training for personnel.(3)Captivity seems to have a negative effect on sperm quality in *Alytes*, but this effect could be reverted. Some differences in the percentage of males responding to hormonal stimulation were observed among captive (33%), wild (50%) and captive males after a semi-free-ranging period (43%). Some captive males that did not respond to hormones released sperm after being allowed to range in the outdoor area. Although sperm samples from captive and wild males (that were in captivity for 16 days before treatment) had similar concentrations and percentages of forward motility, the quality of movement of sperm showing forward progression was different. Sperm in some samples from wild males were moving forward in a straight line and at a faster speed than any sample from captive males. It has been proposed that sperm in terrestrial-fertilizing anuran species are immotile, as found in primitive Leiopelmatidae frogs [33] or in foam-nesting Myobatrachidae [38], and motility may be activated through direct contact with the egg [39]. Further studies will be needed to clarify the factors implicated in sperm activation in *Alytes*. In addition, some samples from captive males that were allowed to range in a semi-free area for a month presented two to four times higher sperm concentration afterward. Similar results were found in *Lithobates chiricahuensis* (Chiricahua leopard frog), with semi-captive and wild males presenting higher sperm concentration than captive ones while no effect on motility was observed [40].

Collection of sperm samples in live amphibians has been performed in various species of several families [1]. Gonadotropin treatment stimulates sperm detachment from the Sertoli cells inside the testis [28]. Then, sperm is transported to the cloaca, and spermic urine can be collected by different methods such as spontaneous urination, cannulation using a catheter or abdominal massage [41]. Catheterization through the cloaca was used as a method for sample collection because it was more reliable than spontaneous urination. However, sometimes males spontaneously urinate and variations in sample characteristics were observed. No sperm was found in a sample obtained by spontaneous urination, although sperm was present after subsequent cannulation of the cloaca. In addition, a sample obtained using the catheter contained 9.8 × 10^5^ spermatozoa/mL while subsequent urination on the petri dish after collection showed 3 × 10^5^ spermatozoa/mL. Among anurans, there are different degrees of association between urinary and reproductive structures. In most species, a connection between both systems exists, which is called the urogenital system, and excretory products and sperm are carried by the same duct to the cloaca. *Alytes* has an extreme condition among amphibians whereby the genital duct is completely separated from the urinary tract, but they merge before entering the cloaca [42,43]. The urinary bladder is connected to the cloaca, and urine moves from the cloaca into the urinary bladder [44] probably by contractions of the cloacal walls [45]. When the cloaca is cannulated, negative pressure is created and urine is drained. Probably, these anatomical characteristics with a separate duct for sperm, together with the compartmentalization between urinary bladder and cloaca, allow the collection of fractions with different sperm concentrations after hormonal stimulation.

In addition to sperm, eggs need to be collected to develop artificial fertilization methods. Stimulation of ovulation was successful and all females released eggs. Although no embryos were obtained from eggs fertilized with stored sperm, 10% of eggs fertilized with recently-collected sperm proceeded up to Gosner stage 21 (when the cornea becomes transparent and the eyes are discernible [46]) and completed embryo development. Considering sperm concentration measured in a sample obtained after AF, the sperm to egg ratio was 700–1000 sperm/egg. Previous studies in other terrestrial-fertilizing anurans found a similar sperm/egg ratio (in *Limnodynastes tasmaniensis*, a foam-nesting species [38]) or even lower (<50 sperm/egg in *Pseudophryne guentheri* [47]) that contrast to the higher ratio observed in aquatic fertilizers (approx. 1–2 × 10^5^ sperm/egg in *Rhinella marina* [48] or *R. arenarum* [49]). Fertilization in terrestrial species becomes more efficient because the probability of sperm–egg encounter is higher since water is not interfering and therefore a decrease in sperm concentration takes place that is usually accompanied by a lower number and a larger size of eggs [50]. Further studies are needed to analyze the effect of assisted reproductive technologies on offspring performance.

## 5. Conclusions

Taken together, our results show that gamete collection by exogenous hormonal stimulation is feasible in males and females, artificial fertilization using these gametes permits production of embryos, and a reliable method for captive maintenance of eggs was implemented. These technologies may contribute to the management of the populations of this and related species.

## Figures and Tables

**Figure 1 animals-13-02802-f001:**
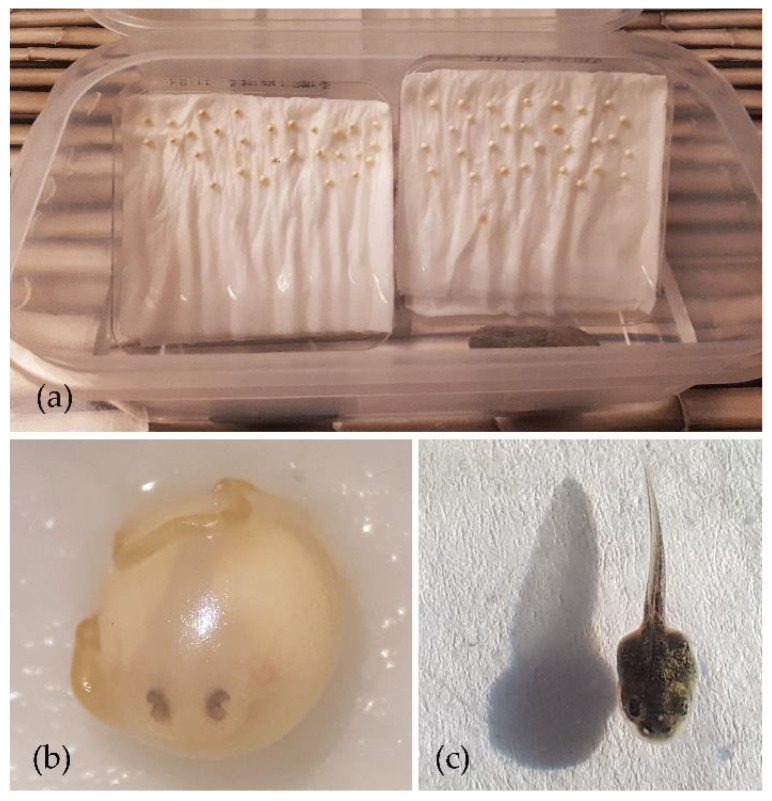
Captive-rearing of embryos and embryo development after artificial fertilization in *Alytes obstetricans*. (**a**) Settings for captive-rearing of embryos, (**b**) seven-day-old embryo presenting optic disc, and (**c**) tadpole 24 h after hatching. Egg diameter = approximately 3 mm.

**Figure 2 animals-13-02802-f002:**
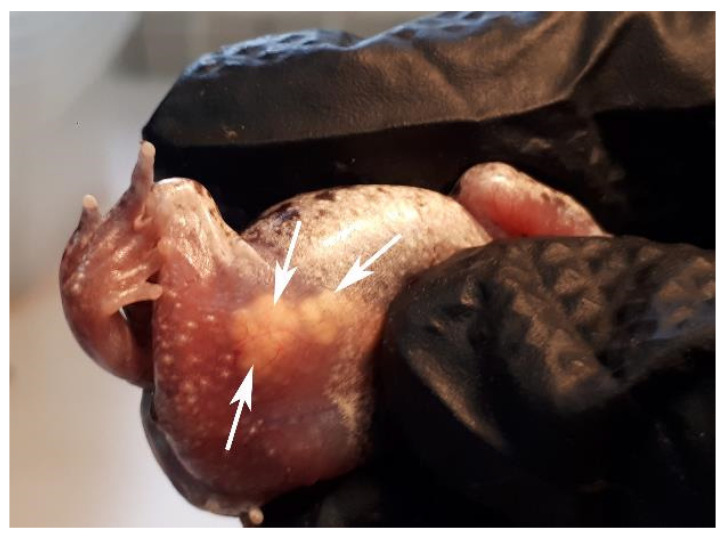
*Alytes obstetricans* female presenting eggs (white arrows) at different developmental stages. In most individuals, eggs can be observed through the bottom part of the abdominal skin.

**Table 1 animals-13-02802-t001:** Number of *Alytes obstetricans* males presenting spermatozoa/total number of males collected in a pilot trial for induction of spermiation using two hormones and two routes of administration.

				Hours After Final Treatment						Total Males *
Priming	Final Treatment	Route	0 h	1 h	2 h	3 h	4 h	5 h	6 h	
none	0.4 µg GnRHa †/g	nasal	NA	0/2	0/2	0/2	0/2	0/2	0/2	0/2
none	0.4 µg GnRHa/g	IP	NA	0/2	0/2	0/2	0/2	0/2	0/2	0/2
none	0.8 µg GnRHa/g	IP	NA	0/4 ‡	0/2 ‡	0/5 ‡	0/3 ‡	0/3 ‡		0/6
none	15 IU hCG/g	IP	NA	0/5 ‡	1/3 ‡	2/6	2/5 ‡	1/2 ‡		2/6
none	0.8 µg GnRHa + 15 IU hCG/g	IP	NA	0/4 ‡	0/3 ‡	2/5 ‡	1/4 ‡	1/3 ‡		2/6
0.03 IU hCG/g	15 IU hCG/g	nasal + IP	0/2	1/2	0/2	0/2				1/2
0.03 IU hCG/g	0.8 µg GnRHa + 15 IU hCG/g	nasal + IP	1/2	1/2	0/2	0/2				1/2
0.025 µg GnRHa/g	15 IU hCG/g	nasal + IP	0/2	0/2	0/2	0/2				0/2
0.025 µg GnRHa/g	0.8 µg GnRHa + 15 IU hCG/g	nasal + IP	0/2	0/2	0/2	0/2				0/2

* Total number of males presenting spermatozoa/total number of males treated, † Gonadotropin-releasing hormone agonist. ‡ Samples were not taken from all treated males at all time points. IP: intraperitoneal. nasal + IP: priming was dispensed via nasal route while final treatment was administered intraperitoneally. 0 h: presence of sperm after the priming doses and before the final hormonal treatment. Priming was administered 24 h before the final hormonal treatment.

**Table 2 animals-13-02802-t002:** *Alytes obstetricans* males presenting sperm/total number of males treated. Response to hormonal induction of spermiation over time from captive and wild animals.

Origin	Route	2 h	4 h	6 h	8 h	10 h	12 h	24 h	Total *
captive	IP	0/12	0/12	2/12	3/12	2/12	3/12	4/12	33%
captive	nasal	0/12	2/12	1/12	1/12	1/12	0/12	0/12	17%
wild	IP	0/4	1/4	1/4	1/4	2/4	2/4	1/4	50%
wild	nasal	0/4	0/4	0/4	0/4	0/4	0/4	0/4	0

Males were treated either with an intraperitoneal (IP) injection of 15 IU hCG/g or with an intranasal dose of 0.04 IU hCG/g, and a urine sample was collected every two hours after hormonal treatment. * Total percentage of responding males.

**Table 3 animals-13-02802-t003:** Sperm concentration (×10^5^ spermatozoa mL^−1^) over time in males presenting sperm after hormonal treatment in captive *Alytes obstetricans* after a semi-free-ranging period. Males were treated with an intraperitoneal injection of either 15 or 20 IU hCG/g.

Animal	Treatment	3 h	6 h	9 h	12 h	24 h
Male 1	15 IU hCG/g	2.2	15.2	6.7	88.0	0
Male 2	15 IU hCG/g	0	0.1	0.3	0.4	0
Male 3	15 IU hCG/g	3.0	9.8	49.9	52.5	30.4
Male 4	20 IU hCG/g	9.1	3.5	6.4	1.5	0
Male 5	20 IU hCG/g	0	0.2	0.4	0	0

## Data Availability

Data are available from the authors upon reasonable request.

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
