# Peer review of "Gamete Collection, Artificial Fertilization and Captive-Rearing of Eggs in a Terrestrial-Breeding Anuran with Parental Care: Alytes obstetricans"

_animals, 2023, doi:10.3390/ani13172802_

Round 1

Reviewer 1 Report

I read and believe that the work "Developing assisted reproductive technologies in a terrestrial breeding anuran with parental care: Alytes obstetricans" has interesting results on the reproduction of A. obstetricans in captivity. However, I found points that I would like to adjust to at work and/or answer.

In Table 1, which I could not understand and is not self-explanatory, in treatments 0.8 μ g GnRHa /g - IP; 15 IU hCG/g – IP and 0.8 μ g GnRHa + 15 IU hCG - IP, 6 males were used in each treatment, but the number of males with the presence of sperm ranged from 0/2 to 2/6. If six males were induced, what is the ratio of 0/2, for example?

Figure 2 is uninformative; I suggest that it be included as supplementary material.

Figure 3 needs to be highlighted or pointed to the eggs with an arrow; with only the photograph, some readers may not identify the eggs.

In table 2, the column "Males with eggs" should be removed or a column for "Second treatment" added as well.

Regarding statistical tests, the Wilcoxon test is a non-parametric hypothesis test used when one wants to compare two paired samples, which is not the case in experiment 1, which, according to the description in the manuscript, could suggest an experiment with a split-plot design, but I believe that because there is not an "A" factor and then a "B" factor, I believe that there are only two treatments, the incubation of them clustered and separated. In this case, the most correct test would be the chi-square and/or the Mann-Whitney U test.

Still, on the statistical tests, in experiment 2, there were "n" enough so that some tests could be performed, leaving the variables of osmolarity, sperm concentration, motility, and other variables of semen quality. Perhaps not to compare application routes, but at least wild versus captive.

In the conclusion, it is stated that "our results show that gamete collection by exogenous hormonal stimulation is feasible in males and females, IVF using these gametes permit production of embryos and a reliable method for captive maintenance of eggs from fertilization until hatching was implemented" However, the results presented do not corroborate the claim that it is a reliable method. In addition, saying that it allows the maintenance of eggs from fertilization to hatching based on three tadpoles is not possible, being almost a case report. However, there are no images or anything to prove the quality with which the tadpoles hatched, neither in experiments 1 nor 4.

Reviewer 2 Report

Developing assisted reproductive technologies in a terrestrial breeding anuran with parental care: Alytes obstetricans

Testing captive-rearing methods and developing assisted reproductive technologies for novel amphibian species is a very important avenue for research. The authors present some interesting findings, however I recommend some major revisions.

Major revisions:

The main experiment rearing embryos together or following separation uses a great experimental design (split-clutch design) and has good replication and appropriate statistics. The experiments developing assisted reproductive technologies, while valuable, often have low replication and data has not been analysed statistically, making the data hard to interpret.

1) I would suggest splitting the paper into two short communications, the first presenting the experiment on rearing embryos and the second presenting the assisted reproductive technology experiments. If the authors choose to keep the experiments in one paper, then the manuscript will need to undergo major revisions to emphasise the embryo experiment and shorten the remaining experiments.

2) Table 1 presenting the pilot study on spermiation would be better presented as supplementary material.

3) Table 3, add a column for the total number of toads releasing sperm in each treatment. The number of responders in each treatment should also be compared statistically.

4) Experiment 5: hormonal stimulation of amplexis has very small samples sizes, animals were reused with only a 2-week gap and no statistics were conducted. For these reasons I suggest removing this experiment from the manuscript. I suggest repeating the study with greater sample sizes and independent replication and publishing as a separate study at a later date.

5) The poor response of males to hormone treatment needs to be emphasised, add the response rates of males to the abstract. Add to the discussion what future studies will be conducted to try to improve the response of males to hormone treatment.

6) Throughout the manuscript please revise English language, particularly spelling, grammar and the use of ‘past’ and ‘present’ tense.

Minor revisions:

1. Title should include mention of testing captive-rearing methods as this was a major part of the study.

2. hyphenate ‘terrestrial-breeding anuran’ in the title

3. lines 11, 12, 25 and throughout. I suggest using consistent terminology, change ‘assisted reproductive techniques’ to ‘assisted reproductive technologies’ or ‘reproductive technologies’. I’ll leave it to the authors to decide which term they prefer, but suggest that the same term is used throughout the manuscript and that these terms aren’t used interchangeably for consistency.

4. line 12, hyphenate ‘terrestrial-breeding anuran’

5. line 13, replace ‘eggs strings’ with ‘egg strings’

6. line 14, replace ‘allowed obtaining sperm and eggs’ with ‘allowed the collection of hormonally-induced sperm and eggs’

7. line 18, replace ‘eggs were maintained separated 100% of clutches had embryo survival’ with ‘eggs were separated, all clutches exhibited embryo survival’

8. line 23, hyphenate ‘terrestrial-breeding species’

9. line 26, replace ‘feasible’ with ‘achieved’

10. line 31, replace ‘clutches (n=13) were removed from males’ with ‘natural clutches (n=13) were removed from wild-caught males’

11. line 24, add the percentage survival in brackets of the one sub-clutch that had embryo survival.

12. In the abstract, the acronyms IVF, hCG and GnRHa need to be written in full at first mention.

13. line 50, replace ‘carries’ with ‘carry’

14. lines 54/55, replace ‘their distribution areas are reduced’ with ‘their area of distribution is also restricted’

15. Throughout the introduction please revise English language, particularly grammar and the use of ‘past’ and ‘present’ tense. The aims paragraph, lines 104-112, changes from past to present tense, please revise.

16. line 117, replace ‘born’ with ‘bred’

17. line 141, replace ‘handling’ with ‘hand’

18. line 195, can you provide the range in room temperature?

19. line 149, figure caption. Replace ‘rear’ with ‘rearing’

20. Table 1, please explain in the caption what the numbers below the collection times represent, is this the ‘number of responders/total animals’?

21. line 258, please present the results of the statistical analysis in full, include the name of the statistical model, f-ratio, degrees of freedom and p-value.

22. line 358 replace ‘sintetic’ with ‘synthetic’

Throughout the manuscript please carefully revise English language, particularly spelling, grammar and the use of ‘past’ and ‘present’ tense.

Reviewer 3 Report

At present, the problem of restoration and maintenance of the abundance of Alytes obstetricans is hampered by both the small size and isolation of populations and the decrease in the genetic diversity, which may be associated with the appearance of deviations in the structure (Goodman et al., 2022). High anthropogenic pressure on habitats, the presence of a complex zone of introgression with different levels of reproductive isolation, require both conservative and active measures to prevent both population reduction and genetic pollution.

The authors paid special attention to the prevention and reduction of fungal infection during the period of development. Set by the authors must be reflected in the conclusion, the implemented areas of research. In the conclusions, it is also necessary to indicate the effect of the presented methodology, including the success of survival and the reproductive properties of the expressed offspring.

Round 2

Reviewer 1 Report

The flaws regarding the statistical tests, and in particular the conclusion, remained and were not satisfactorily answered.

Reviewer 2 Report

Developing assisted reproductive technologies in a terrestrial breeding anuran with parental care: Alytes obstetricans

Testing captive-rearing methods and developing assisted reproductive technologies for novel amphibian species is an important avenue for research. The authors present some interesting findings, however some revisions are still required.

1) Add a column to Table 1 stating the total number of individual responders (across all collection periods) in each hormone treatment.

2) Line 261 and throughout, replace “osmotic pressure” with “osmolality”

3) Experiment 2. Line 262. Add a sentence detailing the overall total number of frogs that responded to IP injection (6/16) and nasal (2/16).

4) Experiment 2. Lines 262-264. I’m not sure that the statistics used are appropriate given the high number of zero (non-responder) values. I suggest removing the description of the statistic in the methods and the following sentence in the results:

 “The model showed no effect of the route of hormone administration (intraperitoneal vs. nasal) on the proportion of responding males in captivity (p>0.05).”

I recommend statistically comparing the number of responding males in each hormone treatment using a contingency table (pearsons chai squared or similar test). You could also use this test to compare the number of responders in the pilot study (table 1)

5) Line 299-300, after the following sentence “The model showed no difference on percentage of responding males between hormone doses (p>0.05).” add that “the lack of statistical difference is likely due to very small sample sizes in this experiment (n=2-3).”

6) The manuscript still has a number of grammatical mistakes throughout, particularly the use of ‘past’ and ‘present’ tense, please edit the manuscript carefully. I will list some of the errors below as examples, but carefully check for others.

- line 14 replace “eggs strings” with “egg strings”

- line 36 replace “eggs string” with “the egg string”

-line 36 replace “eggs split.” with “eggs split individually.”

-line 46 replace “accomplish” with “accomplished”

-line 81 replace “gather” with “gathered”

The manuscript needs some further revision and careful editing to english language
